# Tbx3-dependent amplifying stem cell progeny drives interfollicular epidermal expansion during pregnancy and regeneration

Ryo Ichijo [1,2], Hiroki Kobayashi[1,2], Saori Yoneda[1,2], Yui Iizuka[1,2], Hirokazu Kubo[1,2], Shigeru Matsumura[1,2], Satsuki Kitano[3], Hitoshi Miyachi[3], Tetsuya Honda[4] & Fumiko Toyoshima [1,2]

The skin surface area varies flexibly in response to body shape changes. Skin homeostasis is maintained by stem cells residing in the basal layer of the interfollicular epidermis. However, how the interfollicular epidermal stem cells response to physiological body shape changes remains elusive. Here, we identify a highly proliferative interfollicular epidermal basal cell population in the rapidly expanding abdominal skin of pregnant mice. These cells express Tbx3 that is necessary for their propagation to drive skin expansion. The Tbx3$^+$ basal cells are generated from Axin2$^+$ interfollicular epidermal stem cells through planar-oriented asymmetric or symmetric cell divisions, and express transit-amplifying cell marker CD71. This biased division of Axin2$^+$ interfollicular epidermal stem cells is induced by Sfrp1 and Igfbp2 proteins secreted from dermal cells. The Tbx3$^+$ basal cells promote wound repair, which is enhanced by Sfrp1 and Igfbp2. This study elucidates the interfollicular epidermal stem cell/progeny organisation during pregnancy and suggests its application in regenerative medicine.

[1] Department of Biosystems Science, Institute for Frontier Life and Medical Science, Kyoto University, Sakyo-ku, Kyoto 606-8507, Japan. [2] Department of Mammalian Regulatory Network, Graduate School of Biostudies, Kyoto University, Sakyo-ku, Kyoto 606-8502, Japan. [3] Institute for Frontier Life and Medical Science, Kyoto University, Kyoto 606-8507, Japan. [4] Department of Dermatology, Kyoto University Graduate School of Medicine, Sakyo-ku, Kyoto 606-8501, Japan. Correspondence and requests for materials should be addressed to F.T. (email: ftoyoshi@infront.kyoto-u.ac.jp)

How adult tissue stem cells adapt to physiological changes is a fundamental aspect of stem cell biology. Stem cell self-renewal and differentiation in response to a physiological alteration leads to changes in organ size and tissue homeostasis. Examples include skin, an essential barrier of the body, which alters its surface area flexibly to accommodate body shape changes. However, the processes involved in the epidermal stem cell response to changes in physiological body shape remain unknown.

The epidermis is a stratified epithelium in which basal cells proliferate in the underlying basal layer and eventually move to upper layers to undergo stepwise differentiation[1, 2]. Adult skin maintains epidermal homeostasis by controlling the proliferation and differentiation of stem cells residing in the basal layer of the interfollicular epidermis (IFE)[2]. The classical models that explain IFE homeostasis include the epidermal proliferation unit (EPU) hypothesis that proposes a single slow-cycling stem cell at the centre of each unit divides asymmetrically to give rise to one stem cell and one transit-amplifying (TA) cell progeny that undergoes several rounds of cell division before becoming differentiated cells[3–5]. There is also the stochastic model in which the basal layer consists of a single population of progenitor cells with equivalent potentials for proliferation and differentiation, and their fates are determined stochastically[6–8]. A recent study has proposed a stem cell/committed progenitor hierarchical model, where slow-cycling stem cells generate proliferative committed progenitor cells within the basal layer, which contribute to epidermal homeostasis and regeneration differently[9]. The concept of basal layer heterogeneity is supported by two independent stem cell populations in the skin basal layer[10, 11].

At the onset of stratification of embryonic skin in developing mice, the cell division axis of basal cells shifts from the planar orientation to the basement membrane for a perpendicular orientation, leading to asymmetric cell division that gives rise to a basal undifferentiated cell and suprabasal differentiation-committed cell[12–14]. Therefore, the cell division axis is tightly regulated in embryonic skin to define the self-renewal or asymmetric division of IFE basal cells. In addition to perpendicular asymmetric cell division, adult IFE basal cells undergo planar-oriented asymmetric cell division through which a single basal cell generates one cycling cell and one non-cycling basal cell[6]. Most recent report has demonstrated that a single-basal cell population sustains homeostasis and that planar-oriented divisions are dominant in the basal layer during adult epidermal homeostasis[15]. However, it is unclear how adult IFE stem cells contribute to epidermal tissue reorganisation during changes in physiological body shape.

Here, we demonstrate that in the rapidly expanding abdominal skin of pregnant mice, IFE stem cells undergo planar-oriented asymmetric and symmetric cell divisions to generate highly proliferating cell progeny with distinct cellular properties. These cells express Tbx3 that is necessary for their propagation to drive skin expansion and accommodate foetal growth. We further show that the proteins secreted from dermal cells govern this biased division of the IFE stem cells.

## Results

### Abdominal IFE basal cells proliferate during pregnancy. The abdominal circumference of female C57BL/6N mice was increased slightly by 12 days post-coitus (dpc) and increased drastically between 12 and 16 dpc (Fig. 1a). To determine whether there is an increase in the proliferation of IFE basal cells during pregnancy, we first quantified IFE basal cells stained for the proliferation marker Ki67. In accordance with the increase in abdominal circumference at 12–16 dpc, a population of

Ki67-positive (Ki67$^+$) IFE basal cells had increased gradually and significantly in the abdominal epidermis but not in the dorsal epidermis (Fig. 1b, c). Consistently, basal cells in the abdominal epidermis at 16 dpc incorporated 5-ethynyl-2′-deoxyuridine (EdU) at a significantly higher rate than those in the dorsal epidermis at 16 dpc or in virgin mice (Fig. 1d, e). Furthermore, two-photon microscopic analysis of Fucci2 mice, in which S/G2/M and G1 phase cells can be dual colour imaged by detection of mVenus-hGeminin and mCherry-hCdt1, respectively[16], showed a significant increase in the population of S/G2/M phase cells in the abdominal epidermis at 16 dpc compared with virgin mice (Fig. 1f, g). These results indicate that IFE basal cells in abdominal skin are highly proliferative during pregnancy.

### Tbx3 evokes IFE basal cell proliferation during pregnancy. We performed DNA microarray analysis to identify the genes responsible for the high proliferative potential of IFE basal cells in pregnant mice. By comparing the gene expression profiles of fluorescence-activated cell sorter (FACS)-purified basal epidermal cells (integrin α6$^+$) from abdominal skin of 16 dpc mice with those of abdominal skin of virgin mice and dorsal skin of 16 dpc mice (Supplementary Fig. 1A), we identified 18 candidate genes with >2-fold higher expression levels in the abdominal skin of 16 dpc mice (Supplementary Fig. 1B). Further analysis by quantitative polymerase chain reaction (qPCR) of skin tissues revealed six genes out of the 18 candidates with significantly higher expression levels in the abdominal skin of 16 dpc mice than those in dorsal skin of 16 dpc mice or abdominal skin of virgin mice (Tbx3, Thy1, Tgm3, Stfa1, Stfa3 and Keratin2) (Fig. 2a and Supplementary Fig. 1C). Among them, we focused on Tbx3, a transcription factor that is essential for mammalian development and organogenesis[17, 18]. We detected the expression of Tbx3 protein in IFE basal cells (Fig. 2b), and a population of Tbx3$^+$ IFE basal cells was increased significantly in the abdominal skin of 16 dpc mice compared with virgin mice (Fig. 2c). To assess the requirement of Tbx3 for the proliferation of IFE basal cells in the abdominal skin of pregnant mice, we performed conditional epidermal deletion of Tbx3 using Tbx3$^{flox/flox}$ mice[18] carrying a K14-CreER transgene that is expressed in epidermal basal layers (Tbx3 cKO mice)[19]. Topical tamoxifen treatment of the abdominal and dorsal skin of pregnant mice during 8–16 dpc led to the loss of Tbx3 expression in the IFE basal cells at 16 dpc in Tbx3 cKO mice (Fig. 2d, e, Tbx3 cKO). The loss of Tbx3 resulted in a significant decrease in Ki67$^+$ IFE basal cells at 16 dpc in abdominal skin, whereas it barely altered the Ki67$^+$ IFE basal cell population in dorsal skin (Fig. 2f, g), indicating an essential role of Tbx3 in the proliferation of abdominal IFE basal cells during pregnancy. Notably, a flank of pregnant Tbx3 cKO mice swelled abnormally, which may due to less expansion of abdominal skin (Fig. 2h). Moreover, when the abdominal skin of pregnant mice was cut with a razor, the width of the wound was larger in Tbx3 cKO mice compared with control mice (Fig. 2i, j). In addition, the abdominal epidermis of Tbx3 cKO mice at 16 dpc manifested thinner spinous layers (keratin 10$^+$ layers) (Fig. 2k) and less basal cell density (Fig. 2l) with intense staining signals using an anti-α18-catenin antibody that recognises α-catenin in a force-dependent manner[20, 21] (Fig. 2m). These results indicate that tension applied to the abdominal skin is higher in Tbx3 cKO mice compared with control mice during pregnancy. It is noteworthy that the body weight of embryos from Tbx3 cKO mothers interbred with genetically unaltered wild-type males was slightly decreased compared with that of embryos from control mothers without changes in their gross appearance (Supplementary Fig. 2). Taken together, these results demonstrate that the proliferation of IFE basal cells requires Tbx3 to evoke rapid

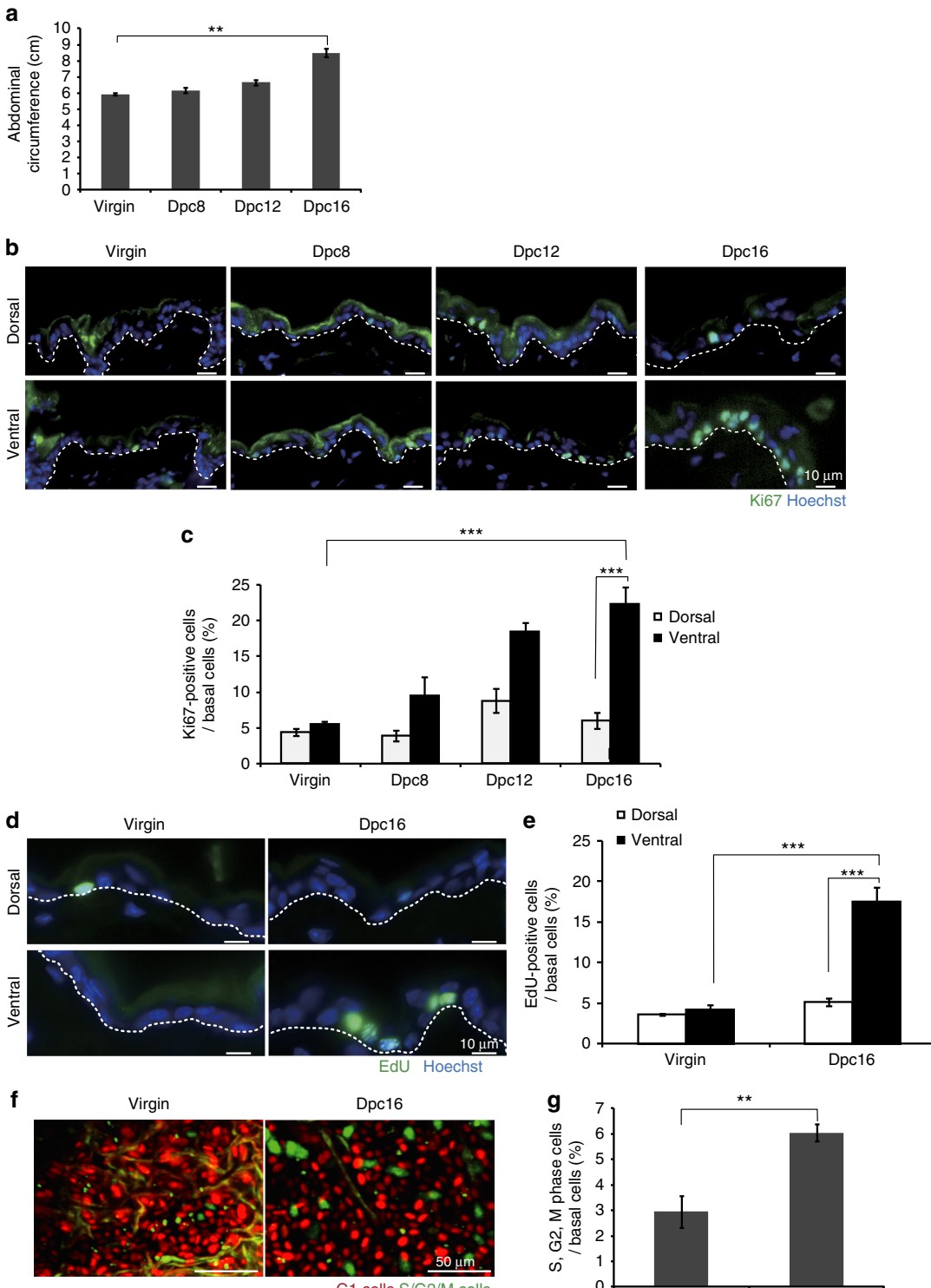

**Fig. 1** IFE basal cells in the abdominal skin of pregnant mice are highly proliferative. **a** Abdominal circumferences of virgin and pregnant (8–16 dpc) mice. Data represent the mean ± standard error of the mean (*s.e.m.*) (*n* = 3 mice); \*\**P* < 0.01, analysed by the two-tailed *t*-test. **b** Immunofluoresence of Ki67 (*green*) and Hoechst (*blue*) on dorsal (*upper*) and ventral (*bottom*) skin from virgin and pregnant (8–16 dpc) mice. **c** Quantification of Ki67$^+$ cells in IFE basal layers. **d** Immunofluoresence of EdU (*green*) and Hoechst (*blue*) on dorsal (*upper*) and ventral (*bottom*) skin from virgin and 16 dpc mice. **e** Quantification of EdU$^+$ cells in IFE basal layers. **f** Two-photon microscopic images of ventral skin in Fucci2 virgin and 16 dpc mice, where cells in G1 and S/G2/M phases exhibit *red* or *green* fluorescence, respectively. **g** Quantification of cells in S/G2/M phases in ventral skin IFE of Fucci2 mice. Data represent the mean ± s.e.m. of averages of seven sections (*n* = 7) pooled from three mice. More than 700 cells were analysed to calculate the average in each section; \*\**P* < 0.01, analysed by the two-tailed *t*-test. **c**, **e** Data represent the mean ± s.e.m. of averages of three independent experiments (*n* = 3). More than 500 cells were analysed to calculate the average in each experiment; \*\**P* < 0.01, \*\*\**P* < 0.001, analysed by Dunnett's multiple comparison test. **b**, **d** *White-dotted lines* indicate basement membranes. Experiments were repeated three times with three mice for representative images in (**b**, **d**, **f**)

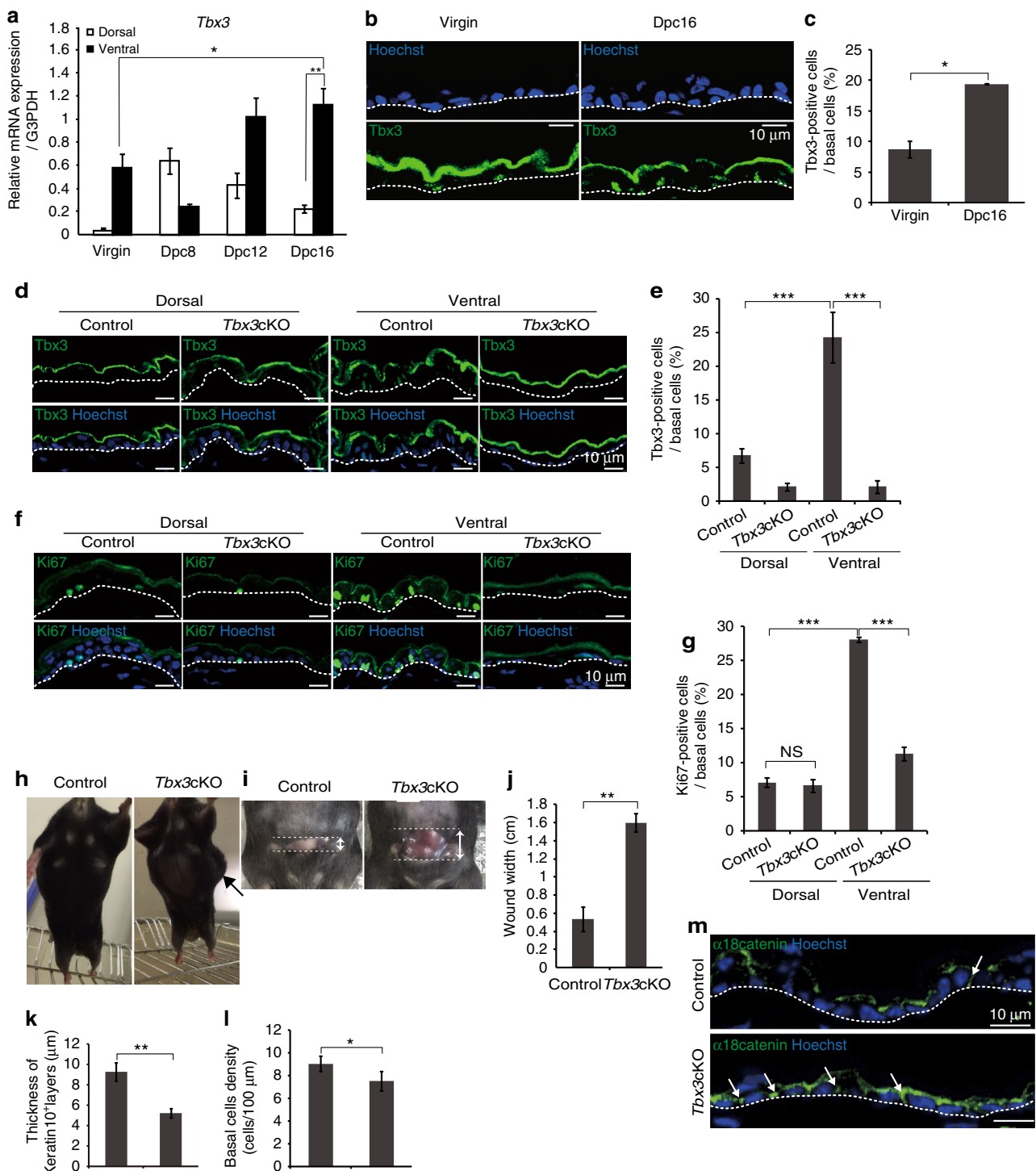

expansion of abdominal skin during pregnancy to deal with foetal growth.

**The Tbx3+ basal cells are Axin2+ IFE stem cell progenies**. We next examined expression of Tbx3 in a distinct subpopulation in the basal layer. A previous report has shown that the Tbx3 protein level peaks at S phase in cultured cells[22], suggesting that Tbx3 marks S phase cells in the basal layer. However, in the abdominal epidermis at 16 dpc, only ~32% of Tbx3+ basal cells ($n = 99$ from three mice) expressed the S/G2/M cell marker geminin, and 54%

of geminin+ cells ($n = 57$ from three mice) expressed Tbx3. In addition, Tbx3 protein was detected in the geminin− cell population (20%, $n = 330$ from three mice), indicating cell cycle-independent expression of Tbx3. Nonetheless, the Tbx3+ IFE basal cells incorporated EdU at a significantly higher rate than Tbx3− IFE basal cells in the abdominal skin at 16 dpc (Fig. 3a, b). These results suggest expression of Tbx3 in a highly proliferative basal cell population that emerges in the abdominal skin during pregnancy.

As a second experiment to characterise the Tbx3+ IFE basal cells, we used Axin2 reporter mice. A recent study has

demonstrated that autocrine Wnt signalling maintains the stemness of IFE basal cells. Thus, the Wnt target gene *Axin2* marks the IFE stem cells that constitute the majority of the basal layer[8]. We used *Axin2-d2EGFP* reporter mice in which expression of a short-lived green fluorescent protein variant (d2EGFP) is driven by the promoter and first intron of the *Axin2* gene[23]. Although we detected d2EGFP signals in only 10% of IFE basal cells in the abdominal skin of *Axin2-d2EGFP* reporter virgin mice, which may be attributed to a short half-life of d2EGFP, we could assess the expression pattern of Axin2-d2EGFP and Tbx3 (Fig. 3c). We found that only ~15% of Axin2-d2EGFP+ IFE basal cells expressed Tbx3 and vice versa in the abdominal skin of pregnant mice (Fig. 3d, e, Dpc16), implicating the largely exclusive expression pattern of each other. In addition, loss of *Tbx3*, which resulted in compromised basal cell proliferation (see Fig. 2g), barely altered the EdU incorporation rate of the Axin2-d2EGFP+ IFE basal cells in the abdominal skin of pregnant mice (Fig. 3f, g, Dpc16). Moreover, in contrast to the Tbx3+ IFE basal cells which increased in number during pregnancy (see Fig. 2c), the fraction of Axin2-d2EGFP+ cells in the basal layer of abdominal skin was significantly decreased during pregnancy, which was reversed by the loss of Tbx3 (Fig. 3h, Dpc16). The EdU incorporation rate of the Axin2-d2EGFP+ IFE basal cells or the fraction of Axin2-d2EGFP+ cells in the basal layer was not altered by the loss of *Tbx3* in virgin mice (Fig. 3g, h, Virgin; Supplementary Fig. 3A). These results suggest that Tbx3 is expressed and propagates a basal cell population that is distinct from the Axin2+ cell pool during pregnancy. However, the EdU incorporation rate of the Axin2-d2EGFP+ IFE basal cells and the population of basal cells that expressed both Axin2-d2EGFP and Tbx3 were increased in the abdominal skin at 16 dpc (see Fig. 3g, d, e), raising the possibility that Axin2 + cells transitioned into Tbx3+ cells. To test this hypothesis, we label-traced *Axin2*-expressing cells using a mouse line carrying an *Axin2-Cre-ERT2* transgene and *H2B-GFP* reporter gene with an upstream floxed stop cassette in the Rosa26 locus[24] (*Axin2-Cre-ER/Rosa26-H2B-GFP* mice). *Axin2-Cre-ER/Rosa26-H2B-GFP* female mice were crossed with wild-type male mice and then treated with tamoxifen at 12 dpc. One day after labelling, only ~5% of Axin2-Cre-ER-labelled cells expressed Tbx3 (Supplementary Fig. 3B, Dpc13), confirming the exclusive expression pattern of Axin2 and Tbx3. However, 4 days after labelling, when the average size and size distribution of Axin2-Cre-ER-labelled clones in the basal layer were increased significantly (Supplementary Fig. 3C, D, Dpc16), >60% of Axin2-Cre-ER-labelled basal cells expressed Tbx3 (Supplementary Fig. 3B, Dpc16), indicating that Axin2+ basal cells transitioned into Tbx3+ basal cells. Intriguingly, 28 days after parturition, Axin2-Cre-ER-labelled basal cells barely expressed Tbx3 (Supplementary Fig. 3B, PD28). In addition, the average size and size distribution of

Axin2-Cre-ER-labelled clones were decreased significantly at postnatal day (PD) 28, whereas they were increased in virgin mice during the same period (Supplementary Fig. 3C, D). These data implied that the majority of Axin2-Cre-ER-labelled cell progeny was eliminated from the basal layer after parturition. Taken together, these results demonstrate that the Axin2+ IFE basal cells give rise to highly proliferative Tbx3+ basal cells during gestation, which are prone to elimination from the basal layer after parturition.

**Axin2+ IFE stem cells undergo planar-asymmetric division**. To further characterise Tbx3+ IFE basal cells, we examined the expression of CD71 (transferrin receptor), a marker of proliferating TA cells[25, 26]. Intriguingly, a fraction of CD71+ IFE basal cells was increased in the abdominal IFE at 16 dpc compared with virgin mice (Fig. 4a, b). In addition, >80% of CD71+ IFE basal cells ($n = 184$ from three mice) expressed Tbx3, and 73% of Tbx3+ IFE basal cells ($n = 204$ from three mice) expressed CD71 (Fig. 4a), whereas only ~7% of Axin2-d2EGFP+ IFE basal cells expressed CD71 and vice versa in the abdominal skin at 16 dpc (Supplementary Fig. 3E; $n > 500$ from three mice). Furthermore, loss of Tbx3 led to a significant decrease in the fraction of CD71+ cells in the basal layer of abdominal skin at 16 dpc (Fig. 4c, d). These results imply that Tbx3 is expressed in the CD71+ cell population and promotes their propagation in the abdominal skin of pregnant mice. Notably, loss of Tbx3 only partially attenuated EdU incorporation into CD71+ IFE basal cells (Supplementary Fig. 3F), suggesting that Tbx3 induces transition from Axin2+ to CD71+ or maintains CD71 expression, but only partially affects the proliferative state of CD71+ cells.

We next determined whether the Axin2+ IFE basal cells are predisposed to generate Tbx3+ basal cells by altering their cell division patterns. To this end, we labelled Axin2+ IFE basal cells by injecting pregnant *Axin2-Cre-ER/Rosa26-H2B-GFP* mice with tamoxifen at 14 dpc and followed their fates after one round of cell division at 16 dpc. During this 48 h labelling period, 1.6 and 5% of the labelled cells had undergone one round of cell division in the dorsal epidermis ($n = 997$ labelled cells from three mice) and abdominal epidermis ($n = 1271$ labelled cells from three mice), respectively, and ~0.5% of the labelled cells were in anaphase/telophase that can be detected by survivin signals at the central spindles (Fig. 4e). First, we quantified the division orientation of Axin2-Cre-ER-labelled IFE basal cells based on the staining pattern of survivin (Fig. 4e, *upper panels*). We found that >85% of the labelled cells had divided parallel to the basement membrane in both dorsal and abdominal skin at 16 dpc (Fig. 4e), indicating that Axin2-Cre-ER-labelled IFE basal cells preferentially divide parallel to the basement membrane. However, in the survivin− paired cell population, which had already exited mitosis and entered interphase of the next cell cycle, the

**Fig. 2** Tbx3 is required for proliferation of IFE basal cells and abdominal skin expansion during pregnancy. **a** qPCR analysis of *Tbx3* gene expression in dorsal and ventral skin from virgin and pregnant (8–16 dpc) mice. Data represent the mean ± s.e.m. ($n = 3$ mice); *$P < 0.05$, **$P < 0.01$, analysed by Dunnett's multiple comparison test. **b** Immunofluoresence of Tbx3 (*green*) and Hoechst (*blue*) on ventral skin from virgin and 16 dpc mice. **c** Quantification of Tbx3+ cells in IFE basal layers. **d, f** Immunofluoresence of Tbx3 (**d**) or Ki67 (**f**) (*green*) and Hoechst (*blue*) on tamoxifen-treated dorsal and ventral skin of 16 dpc control (*Tbx3*flox/flox/*WT*) and *Tbx3* cKO (*Tbx3*flox/flox/*K14-CreER*) mice interbred with genetically unaltered males. **e, g** Quantification of Tbx3+ cells (**e**) and Ki67+ cells (**g**) in IFE basal layers of dorsal and ventral skin from 16 dpc mice. **h** Representative images of 16 dpc control and *Tbx3* cKO mice. An *arrow* indicates a swollen flank. **i** Representative images of razor-cut abdominal skin of 16 dpc control and *Tbx3* cKO mice. The skin was laterally cut with a razor at 1.5 cm in length. *Arrows* indicate the wound width. **j** Quantification of the wound width. **k** Thickness of keratin 10+ layers of ventral skin from 16 dpc control and *Tbx3* cKO mice. **l** IFE basal cell density in ventral skin from 16 dpc control and *Tbx3* cKO mice. **m** Immunofluoresence of α18-catenin (*green*) and Hoechst (*blue*) on ventral skin from 16 dpc control and *Tbx3* cKO mice. **c, g** Data represent the mean ± s.e.m. of averages of three independent experiments ($n = 3$). > 500 cells (**c**), > 220 cells (**e**), > 200 cells (**g**) were analysed to calculate the average in each experiment; **$P < 0.01$, ***$P < 0.001$, analysed by the two-tailed *t*-test (**c**), Dunnett's multiple comparison test (**e**) or Tukey's multiple comparison test (**g**). **j–l** Data represent the mean ± s.e.m. ($n = 3$ mice); *$P < 0.05$, **$P < 0.01$, analysed by the two-tailed *t*-test. **b, m** *White-dotted lines* indicate basement membranes. Experiments were repeated three times with three mice for representative images in (**b, d, f, h, i, m**)

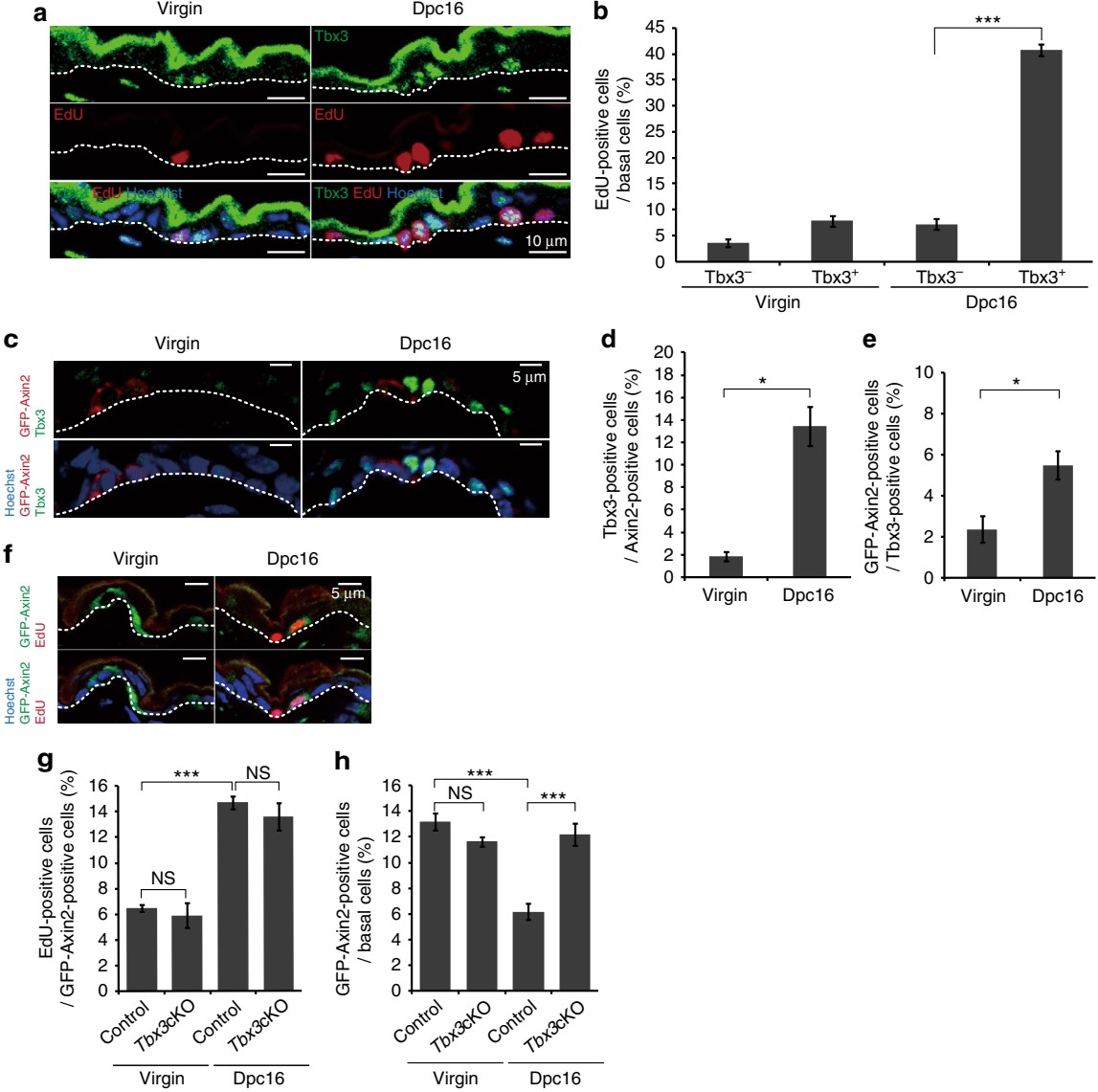

**Fig. 3** Axin2+ IFE stem cells give rise to highly proliferative Tbx3+ basal cells during pregnancy. **a** Immunofluoresence of Tbx3 (*green*), EdU (*red*) and Hoechst (*blue*) on ventral skin from virgin and 16 dpc mice. **b** Quantification of EdU+ cells in IFE basal layers. **c** Immunofluoresence of Tbx3 (*green*), GFP (*red*) and Hoechst (*blue*) on ventral skin from virgin and 16 dpc *Axin2-d2EGFP* mice. **d**, **e** Quantification of Tbx3+ cells in the GFP-Axin2+ cell population (**d**) and GFP-Axin2+ cells in the Tbx3+ cell population (**e**) in IFE basal layers. **f** Immunofluoresence of GFP (*green*), EdU (*red*) and Hoechst (*blue*) on ventral skin from virgin and 16 dpc *Axin2-d2EGFP* mice. **g** Quantification of EdU+ cells in the GFP-Axin2+ cell population in IFE basal layers on ventral skin of virgin and 16 dpc *Axin2-d2EGFP* mice and *Axin2-d2EGFP/Tbx3* cKO mice. **h** Quantification of GFP-Axin2+ cells in IFE basal layers. **b**, **d**, **e**, **g**, **h** Data represent the mean ± s.e.m. of averages of three independent experiments (*n* = 3). > 500 cells (**b**, **h**), > 50 cells (**d**, **e**), > 450 cells (**g**) were analysed to calculate the average in each experiment; *$P < 0.05$, ***$P < 0.001$, analysed by the two-tailed *t*-test (**b**, **d**, **e**) or Tukey's multiple comparison test (**g**, **h**). **a**, **c**, **f** *White-dotted lines* represent basement membranes. Experiments were repeated three times with three mice for representative images

number of two basal cell pairs was significantly decreased in the dorsal epidermis compared with the abdominal epidermis at 16 dpc (Fig. 4f). These results indicate that Axin2-Cre-ER-labelled IFE basal cells divide parallel to the basement membrane followed by upward differentiation of one daughter cell in the dorsal epidermis, whereas both daughter cells remained attached to the basement membrane after cell division in abdominal skin at 16 dpc. Next, we examined the expression patterns of Tbx3 in the divided Axin2-Cre-ER-labelled IFE basal cells. In the dorsal epidermis, the divided Axin2-Cre-ER-labelled IFE basal cells had generated either two basal cells or one basal/one suprabasal cell, and their daughter cells rarely expressed Tbx3 (Fig. 4g, patterns 1 and 2). Conversely, in the abdominal epidermis, the majority of the divided Axin2-Cre-ER-labelled IFE basal cells had generated

two basal cells with symmetric or asymmetric expression of Tbx3 (Fig. 4g, h; patterns 3 and 4). In pattern 3, >90% of the paired cells expressed CD71 in both daughter cells (Supplementary Fig. 3G; *n* = 21 from three mice), confirming that pattern 3 represents symmetric division of an Axin2-Cre-ER-labelled IFE basal cell that gives rise to a pair of Tbx3+/CD71+ cells. As a second approach to investigate the cell division pattern during pregnancy, we labelled epidermal basal cells at clonal density by injecting pregnant *K14-Cre-ERT2*[27] */R26-H2B-GFP* mice with a very low dose of tamoxifen (0.2 mg/25 g body weight). In the abdominal epidermis of virgin mice, where only a small fraction of IFE basal cells underwent cell division (see Fig. 1c), the majority of these cells had generated either two basal cells or one basal/one suprabasal cell without expressing Tbx3

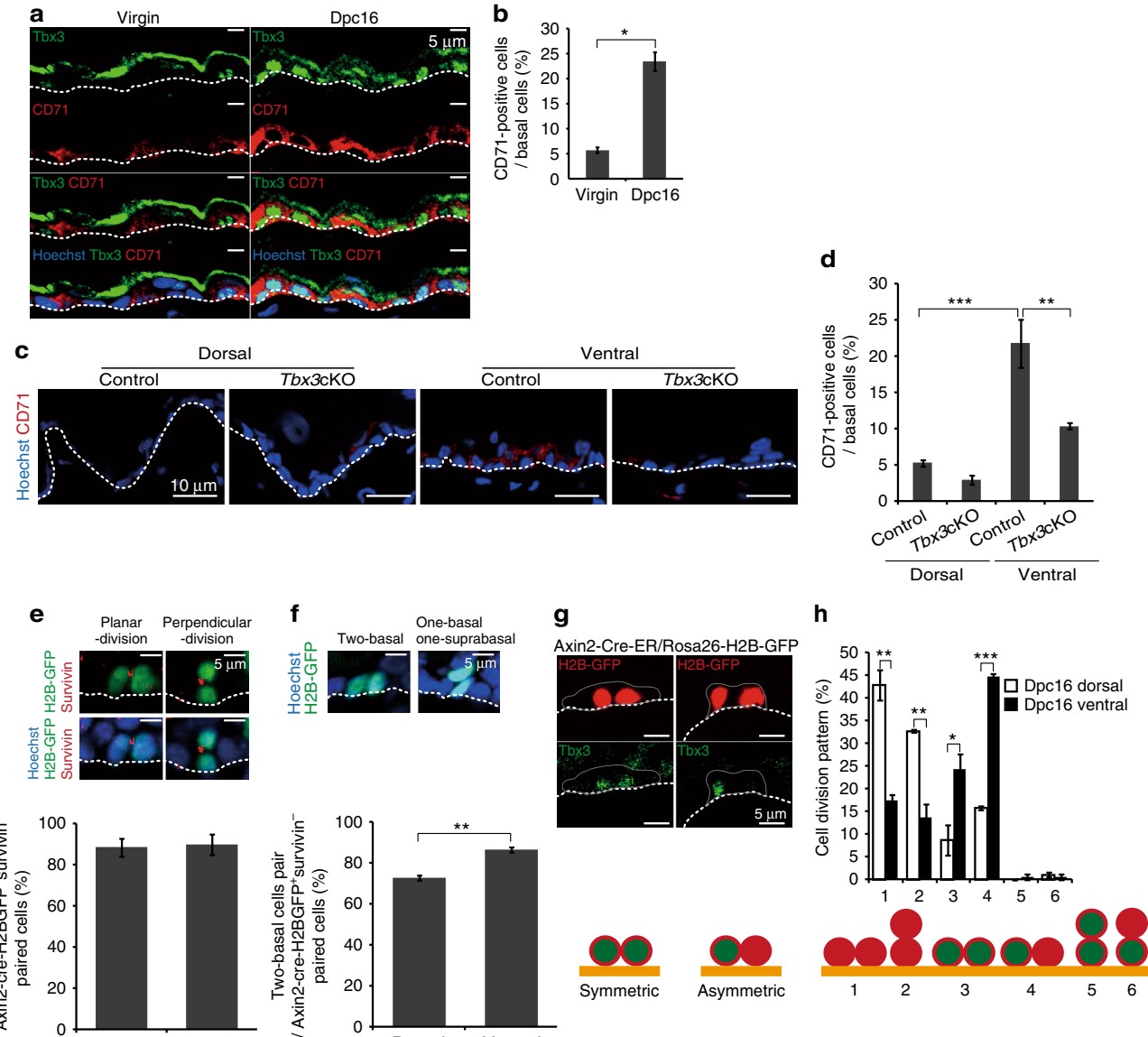

**Fig. 4** Axin2[+] IFE stem cells generate Tbx3[+] basal cells through planar-oriented symmetric/asymmetric cell division. **a** Immunofluoresence of Tbx3 (*green*), CD71 (*red*) and Hoechst (*blue*) on ventral skin from virgin and 16 dpc mice. **b** Quantification of CD71[+] cells in IFE basal layers. **c** Immunofluoresence of CD71 (*red*) and Hoechst (*blue*) on tamoxifen-treated dorsal and ventral skin from 16 dpc control and Tbx3 cKO mice interbred with genetically unaltered males. **d** Quantification of CD71[+] cells in IFE basal layers. **e, f** Immunofluoresence of GFP (*green*), survivin (*red*) and Hoechst (*blue*) on ventral skin from 16 dpc Axin2-Cre-ER/Rosa26-H2B-GFP mice. Quantification of planar-oriented IFE basal cell division in Axin2-Cre-ER labelled-survivin[+] cells (**e**) and the two basal cell pairs in Axin2-Cre-ER labelled-survivin[−] cells (**f**) are shown on the *bottom*. **g** Typical images of planar-oriented symmetric and asymmetric division of Axin2-Cre-ER-labelled IFE basal cells. **h** Quantification of the division pattern of Axin2-Cre-ER-labelled IFE basal cells in dorsal and ventral skin at 16 dpc. **b, d–f, h** Data represent the mean ± s.e.m. of averages of three independent experiments (*n* = 3). > 500 cells (**b**), > 200 cells (**d**), > 40 cells (**e**), > 150 cells (**f**), > 50 cells (**h**) were analysed to calculate the average in each experiment; *P < 0.05, **P < 0.01, ***P < 0.001, analysed by the Dunnett's multiple comparison test (**d**), two-tailed *t*-test (**b, e, f**), or Tukey's multiple comparison test (**h**). **a, c, e, f, g** *White-dotted lines* represent basement membranes. Experiments were repeated three times with three mice for representative images

(Supplementary Fig. 3H, I; patterns 1 and 2). Conversely, the abdominal epidermis of pregnant mice at 16 dpc displayed a significant increase in the population of IFE basal cells that divided parallel to the basement membrane by symmetric or asymmetric cell division to gives rise to one or two Tbx3[+] basal cells (Supplementary Fig. 3H, I; patterns 3 and 4). The proportion of parallel symmetric cell division (pattern 3) was higher in K14-Cre-ER-labelled IFE basal cells compared with Axin2-Cre-ER-labelled IFE basal cells (see Fig. 4h), implying planar cell division of Tbx3[+] (Axin2[−]) IFE basal cells. Next, we examined the cell division orientation of IFE basal cells by staining for survivin. We

found that planar-oriented basal cell division was significantly increased in the abdominal epidermis compared with the dorsal epidermis at 16 dpc (Supplementary Fig. 3J). In addition, loss of Tbx3 attenuated the planar-oriented basal cell division significantly in the abdominal epidermis, but only slightly in the dorsal epidermis at 16 dpc, indicating that the Tbx3[+] IFE basal cells emerging in abdominal skin of pregnant mice are predisposed to divide parallel to the basement membrane. Taken together, these results demonstrate that the Axin2[+] IFE basal cells in abdominal skin activate during pregnancy and undergo planar-oriented asymmetric or symmetric cell division to give rise to the Tbx3[+]

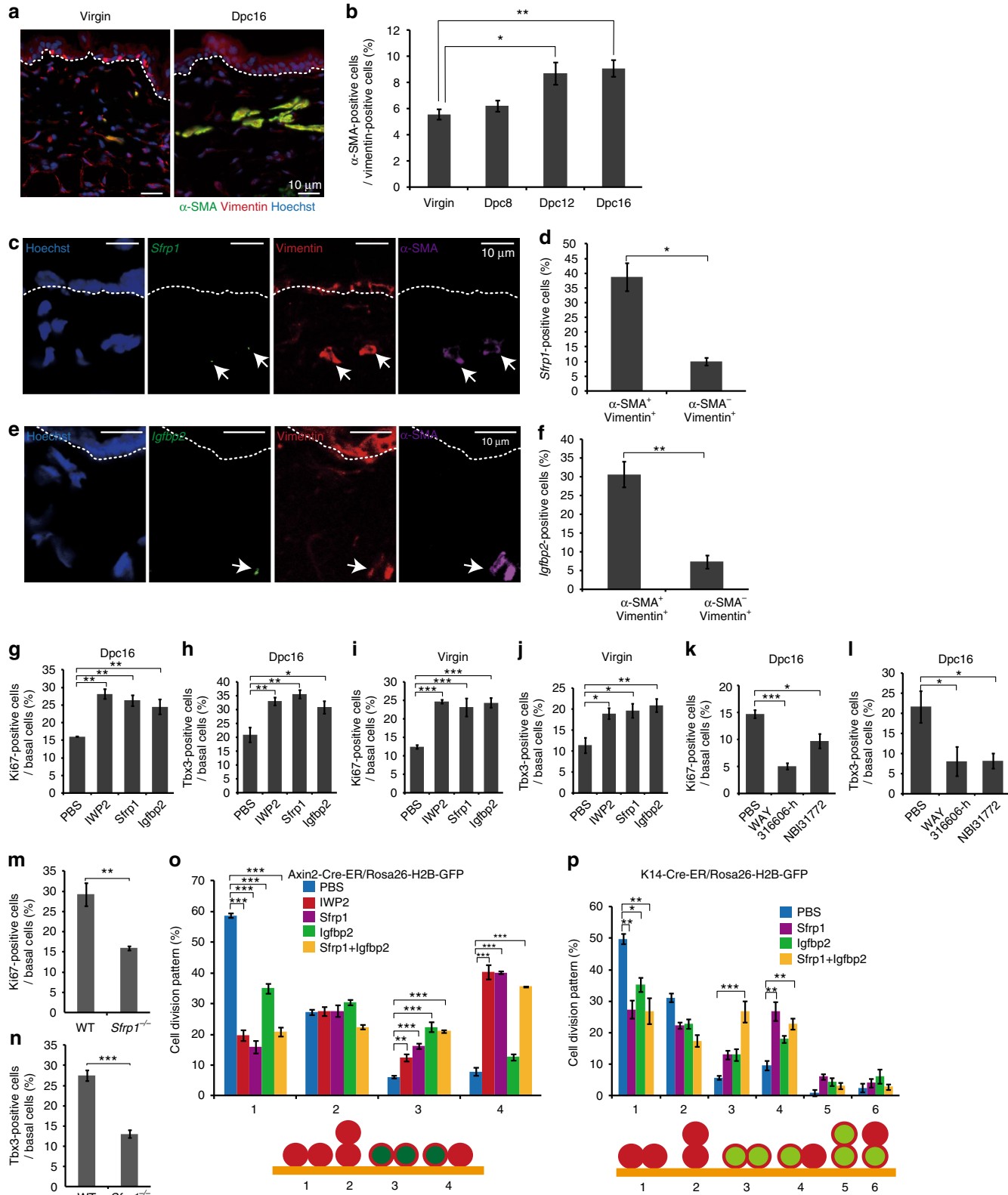

basal cells. These Tbx3[+] basal cells proliferate and divide preferentially parallel to the basement membrane, leading to the epithelial expansion of abdominal skin during pregnancy.

**Dermal proteins govern IFE stem/progeny cell dynamics.** Next, we determined the source of the signals that regulate the

dynamicity of the Axin2[+] and Tbx3[+] IFE basal cells in the abdominal skin of pregnant mice. Increasing evidence has suggested cross-talk between dermal and epidermal cells in normal and malignant skin tissues[28, 29]. Therefore, we hypothesised that secretory proteins derived from dermal cells are the source of the signals for Axin2[+] and Tbx3[+] IFE basal cells in the abdominal skin of pregnant mice. In support of this hypothesis, a population

of α-smooth muscle actin (α-SMA)$^+$/vimentin$^+$ cells, which are known to emerge during tissue regeneration[30], was increased significantly in the dermis of abdominal skin during the course of gestation (Fig. 5a, b). Intradermal injection of either PP2 or LY364947, inhibitors of Src and transforming growth factor-β type I receptor respectively, significantly reduced the population of α-SMA$^+$/vimentin$^+$ in the dermis as reported previously[31, 32] (Supplementary Fig. 4A, D), further confirming the increase of a α-SMA$^+$/vimentin$^+$ cell population in the dermis of abdominal skin during pregnancy. Notably, Ki67$^+$ IFE basal cells were decreased significantly in abdominal skin of pregnant mice by intradermal injection of PP2 or LY364947 (Supplementary Fig. 4B, E). Moreover, Tbx3 mRNA expression and the population of Tbx3$^+$ IFE basal cells in the abdominal skin were decreased significantly in these mice (Supplementary Fig. 4C, F). These results prompted us to identify the dermal-derived secreted proteins that induce the emergence of Tbx3$^+$ IFE basal cells in the abdominal skin of pregnant mice.

To this end, we performed DNA microarray analysis to identify genes expressed predominantly in the dermis of pregnant mice. By comparing the gene expression profiles of dermal tissues from abdominal skin of 15 dpc mice with abdominal skin of virgin mice, abdominal skin of PP2-injected 15 dpc mice, and dorsal skin of 15 dpc mice, we identified 129 candidate genes with >2-fold higher expression levels in the abdominal dermis of 15 dpc mice (Supplementary Fig. 4G, H). Seven genes out of the 129 candidates encode secretory proteins, including secreted frizzled-related protein 1 (Sfrp1), a soluble protein that downregulates Wnt signalling by directly binding to Wnt proteins and Fz receptors[33] (Supplementary Fig. 4I). Fluorescence in situ hybridisation (FISH) in combination with α-SMA immunofluorescence confirmed the expression of sfrp1 mRNA in α-SMA$^+$ cells residing in the dermis (Supplementary Fig. 4J). In virgin mice, ~30% of α-SMA$^+$ dermal cells in the abdominal skin expressed sfrp1 mRNA, which was slightly increased (~40%) in pregnant mice at 16 dpc (Supplementary Fig. 4K). The expression of sfrp1 mRNA was enriched in the α-SMA$^+$/vimentin$^+$ cell population compared with the α-SMA$^-$/vimentin$^+$ cell population residing in the dermis (Fig. 5c, d). No signal of sfrp1 mRNA was detected in the epidermis ($n > 900$ basal cells from three mice). To assess the contribution of Sfrp1 to IFE basal cell dynamics, we intradermally injected the abdominal skin of pregnant or virgin mice with recombinant Sfrp1 protein. The results showed that injection of Sfrp1 protein significantly increased the populations of Ki67$^+$ and Tbx3$^+$ IFE basal cells in both pregnant and virgin mice (Fig. 5g–j).

Similar to Sfrp1 protein, intradermal injection of IWP-2, a validated small molecule inhibitor of Wnt secretion[34], significantly increased the populations of Ki67$^+$ and Tbx3$^+$ IFE basal cells in the abdominal skin of both pregnant and virgin mice (Fig. 5g–j), indicating that Wnt inhibitory signals function as a trigger for the emergence of Tbx3$^+$ cells in the IFE basal layer.

Conversely, wild-type pregnant mice injected intradermally with WAY 316606-h, an inhibitor of Sfrp1[35], manifested a significant decrease in the population of Ki67$^+$ and Tbx3$^+$ IFE basal cells in their abdominal skin (Fig. 5k, l). In addition, a similar phenotype was observed in Sfrp1$^{-/-}$[36] pregnant mice interbred with wild-type males (Fig. 5m, n), confirming an essential role of Sfrp1 in the induction of Tbx3$^+$ IFE basal cells during pregnancy. To further assess the roles of Sfrp1 in the dynamics of IFE basal cells, recombinant Sfrp1 protein was intradermally injected into the abdominal skin of virgin Axin2-Cre-ER/Rosa26-H2B-GFP and K14-Cre-ER/R26-H2B-GFP mice, and the division patterns of Axin2-Cre-ER-labelled IFE basal cells and K14-Cre-ER-labelled IFE basal cells were examined, respectively (Fig. 5o, p). Similar to pregnant mice, injection of either Sfrp1 protein or IWP2 into virgin mice induced the planar-oriented asymmetric or symmetric division of Axin2-Cre-ER-labelled IFE basal cells to give rise to a single or pair of Tbx3$^+$ basal cells, respectively (Fig. 5o; patterns 3 and 4). In addition, injection of Sfrp1 protein significantly increased the population of K14-Cre-ER-labelled IFE basal cells that asymmetrically divided parallel to the basement membrane to give rise to Tbx3$^+$ and Tbx3$^-$ daughter cells (Fig. 5p; pattern 4), indicating an essential role of Sfrp1 in the regulation of IFE basal cell division patterns. However, unlike pregnant mice, injection of Sfrp1 protein into virgin mice only partially induced the planar symmetric division of K14-Cre-ER-labelled IFE basal cells (Fig. 5p; pattern 3), suggesting the existence of additional dermal-derived signals in pregnant mice. Our DNA microarray analysis identified insulin-like growth factor-binding protein 2 (Igfbp2) as another gene upregulated in the dermis of pregnant mice (see Supplementary Fig. 4I). We found Igfbp2 mRNA in α-SMA$^+$ cells residing in the dermis, and the population of Igfbp2 mRNA-expressing α-SMA$^+$ cells was slightly increased in the abdominal skin of pregnant mice (Supplementary Fig. 4L, M). We also observed enriched expression of Igfbp2 mRNA in the α-SMA$^+$/vimentin$^+$ cell population compared with the α-SMA$^-$/vimentin$^+$ cell population residing in the dermis (Fig. 5e, f). No signal of Igfbp2 mRNA was detected in the epidermis ($n > 900$ basal cells from three mice). In addition, similar to Sfrp1, injection of recombinant Igfbp2 protein significantly increased the populations of Ki67$^+$ and Tbx3$^+$ IFE basal cells in both pregnant and virgin mice (Fig. 5g–j). Moreover, injection of wild-type pregnant mice with NBI31772, an inhibitor of Igfpb2[37], significantly decreased the population of Ki67$^+$ and Tbx3$^+$ IFE basal cells in their abdominal skin (Fig. 5k, l). Notably, unlike Sfrp1, intradermal injection of Igfbp2 protein into virgin mice did not induce planar-oriented asymmetric division of the Axin2-Cre-ER-labelled IFE basal cells, although it induced planar-oriented symmetric division to generate two Tbx3$^+$ basal cells (Fig. 5o; patterns 3 and 4). Furthermore, Igfbp2 together with Sfrp1, but not either protein alone, significantly induced planar-oriented symmetric division of

Fig. 5 Dermal-derived Sfrp1 and Igfbp2 proteins induce the emergence of IFE Tbx3$^+$ basal cells. **a** Immunofluoresence of α-SMA (*green*), vimentin (*red*) and Hoechst (*blue*) on ventral skin from virgin and 16 dpc mice. **b** Quantification of α-SMA$^+$ cells in the vimentin$^+$ cell population in the dermis. **c, e** FISH analysis of Sfrp1 and Igfbp2 mRNAs with anti-sense probes (*green*) in combination with immunofluoresence of α-SMA (*magenta*), vimentin (*red*) and Hoechst (*blue*) on ventral skin from 16 dpc mice. **d, f** Quantification of Sfrp1$^+$ (**d**) or Igfbp2$^+$ (**f**) cells in the α-SMA$^+$/vimentin$^+$ or α-SMA$^-$/vimentin$^+$ cell populations in the dermis. **g–j** Quantification of Ki67$^+$ (**g, i**) and Tbx3$^+$ (**h, j**) cells in IFE basal layers of ventral skin from 16 dpc mice (**g, h**) and virgin mice (**i, j**) injected with PBS, IWP2, Sfrp1 or Igfbp2. **k, l** Quantification of Ki67$^+$ and Tbx3$^+$ cells in IFE basal layers of ventral skin from 16 dpc mice injected with PBS, WAY316606-h or NBI31772. **m, n** Quantification of Ki67$^+$ (**m**) and Tbx3$^+$ (**n**) cells in basal layers of ventral skin from 16 dpc wild-type mice and 16 dpc Sfrp1$^{-/-}$ mice interbred with wild-type males. **o, p** Quantification of the division pattern of Axin2-Cre-ER-labelled IFE basal cells (**o**) and K14-Cre-ER-labelled IFE basal cells (**p**) in ventral skin from virgin mice injected with PBS, IWP2, Sfrp1, Igfbp2 or Sfrp1 + Igfbp2. **b, d, f–p** Data represent the mean ± s.e.m. of averages of three independent experiments ($n = 3$). >100 cells (**b**), >140 cells (**d**), >150 cells (**f**), >500 cells (**g–n**), >50 cells (**o, p**) were analysed to calculate the average in each experiment; *$P < 0.05$, **$P < 0.01$, ***$P < 0.001$, analysed by Dunnett's multiple comparison test (**b, g–l**), the two-tailed *t*-test (**d, f, m, n**) or Tukey's multiple comparison test (**o, p**). **a, c, e** *White-dotted lines* represent basement membranes. Experiments were repeated three times with three mice for representative images

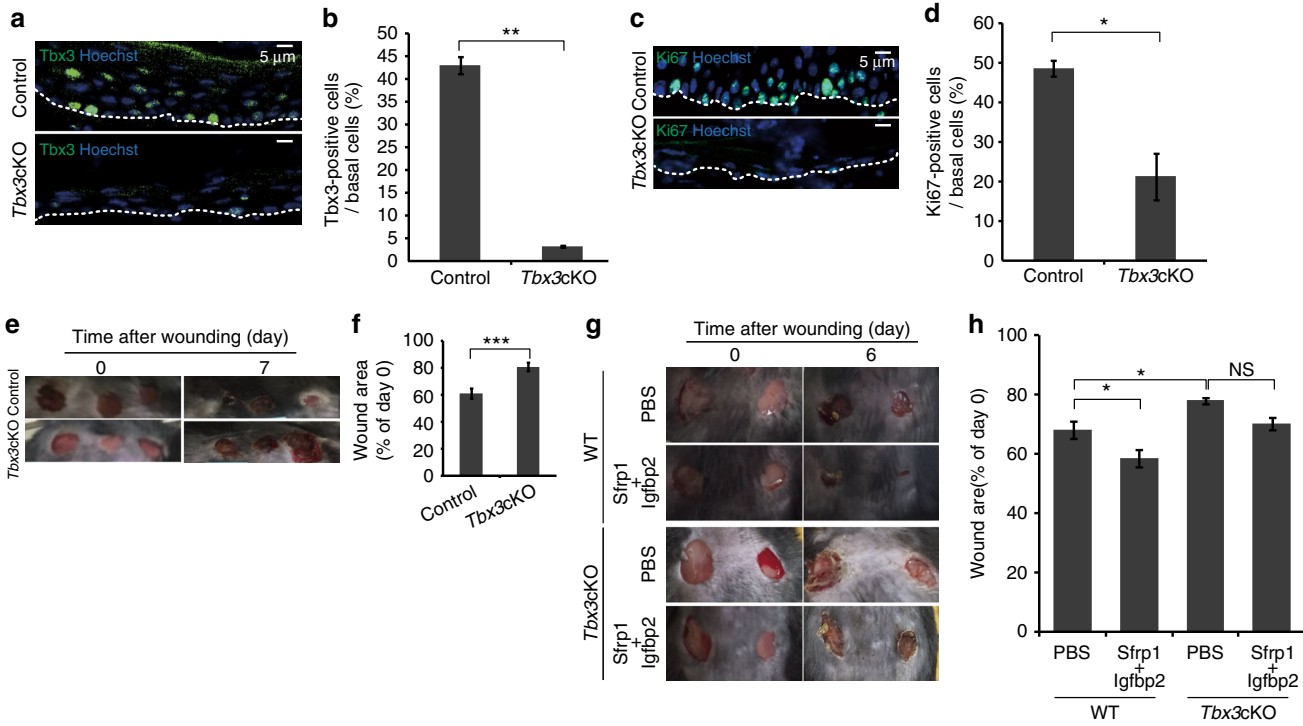

**Fig. 6** Topical administration of Sfrp1 and Igfbp2 proteins facilitate tissue repair upon wounding. **a, c** Immunofluoresence of Tbx3 (**a**) or Ki67 (**c**) (*green*) and Hoechst (*blue*) on dorsal skin from control and *Tbx3* cKO mice at 1-week post-wounding. **b, d** Quantification of Tbx3+ cells (**b**) and Ki67+ cells (**d**) in IFE basal layers from dorsal skin at 1-week post-wounding. **e** Representative images of dorsal skin of control and *Tbx3* cKO mice at days 0 and 7 post-wounding. **f** Quantification of wound healing. The diameter of each wound site was measured, and the average diameter at day 7 post-wounding relative to that at day 0 is shown. **g** Representative images of dorsal skin of wild-type and *Tbx3* cKO mice at day 6 post-wounding injected with or without Sfrp1 and Igfbp2 proteins. The proteins were injected once per day for 4 days after induction of wounding. **h** Quantification of wound healing. The average diameter of each wound site at day 6 post-wounding relative to that at day 0 is shown. **b, d** Data represent the mean ± s.e.m. of averages of three independent experiments ($n=3$). More than 500 cells were analysed to calculate the average in each experiment; *$P<0.05$, **$P<0.01$, analysed by the two-tailed *t*-test. **f, h** Data represent the mean ± s.e.m. of averages of twelve wound sites ($n=12$) pooled from three mice; *$P<0.001$, ***$P<0.001$ analysed by the two-tailed *t*-test (**f**) or Tukey's multiple comparison test (**h**). **a, c, e, g** *White-dotted lines* represent basement membranes. Experiments were repeated three times with three mice for representative images

K14-Cre-ER-labelled IFE basal cells to give rise to two Tbx3+ basal cells (Fig. 5p; patterns 3), which recapitulated the cell division pattern of K14-Cre-ER-labelled IFE basal cells in the abdominal skin of pregnant mice (see Supplementary Fig. 3I). Therefore, both Sfrp1 and Igfbp2 proteins govern the IFE basal cell dynamicity in the abdominal skin of pregnant mice.

**The Tbx3+ basal cells promote wound healing.** Stem cells in various organs activate upon injury to regenerate damaged tissue. Therefore, we determined whether Tbx3 IFE basal cells play a role in skin tissue repair after injury by inducing punch wounds through full thickness skin biopsies in the dorsal skin. We found an increase in the population of Tbx3+ IFE basal cells in wounded sites, which were diminished in *Tbx3* cKO mice (Fig. 6a, b). In addition, the wound-evoked Ki67+ proliferative IFE basal cell population was attenuated in *Tbx3* cKO mice (Fig. 6c, d), resulting in a significant delay in wound healing after injury in *Tbx3* cKO mice compared with control mice (Fig. 6e, f). These results indicate that Tbx3+ cells play a pivotal role in skin tissue repair after injury. Because injection of recombinant Sfrp1 and Igfbp2 proteins increased the population of Tbx3+ IFE basal cells in virgin mice (see Fig. 5i, j), we assessed the latent potential of these proteins to facilitate tissue repair upon wounding. The results showed that topical administration of Sfrp1 and Igfbp2 proteins on wounding site significantly increased the healing rate after injury in wild-type mice. This healing effect of these proteins was attenuated in the *Tbx3* cKO epidermis (Fig. 6g, h). These

results demonstrate that Sfrp1 and Igfbp2 proteins facilitate tissue repair upon wounding via propagation of the Tbx3+ basal cell population.

**Discussion**

In this study, we have revealed the organisation of IFE basal layers in the rapidly expanding abdominal skin of pregnant mice. Based on our results, we propose the following model. During the course of gestation, α-SMA+/vimentin+ dermal cells secrete Sfrp1 and Igfpb2. These proteins function as a trigger for the planar-oriented asymmetric or symmetric cell division of Axin2+ IFE basal cells to generate highly proliferative Tbx3+ cells within the basal layer. The Tbx3+ IFE basal cells further divide parallel to the basement membrane to drive abdominal skin expansion during pregnancy (Supplementary Fig. 5A).

How does this model relate to the postulated models of IFE homeostasis? According to the classical EPU hypothesis, slow-cycling IFE stem cells divide with invariant asymmetry to generate one stem and one TA cell. Axin2+ IFE basal cells are capable dividing either asymmetrically or symmetrically to give rise to one or two Tbx3+ basal cells. This result supports the previous notion that Axin2+ cells are representative of the general population of IFE basal cells with an unbiased fate choice[8], although we cannot exclude the possibility that a subpopulation of Axin2+ IFE basal cells undergoes biased asymmetric cell division. We have not determined whether Tbx3+ IFE basal cells are fate-committed progenitor cells, which undergo terminal

differentiation after several rounds of division, or capable of generating Axin2$^+$ IFE basal cells reversibly. Because the majority of Axin2$^+$ IFE basal cell progenies, including Tbx3$^+$ IFE basal cells, are eventually eliminated from the basal layer after parturition, Tbx3$^+$ basal cells are either committed to undergo elimination or reversion to the Axin2$^+$ state for elimination stochastically. In either case, our results provide evidence that IFE stem cells are capable generating highly proliferating cell progeny with distinct cellular properties during changes in physiological body shape.

Our data show that dermal-derived secreted proteins Sfrp1 and Igfbp2 function as a trigger for the emergence of Tbx3$^+$ IFE basal cells through symmetric or asymmetric division of Axin2$^+$ IFE basal cells. Because IFE stem cells are maintained by autocrine Wnt signalling[8], Sfrp1 may directly repress autocrine Wnt signalling to induce Axin2$^+$-to-Tbx3$^+$ cell transition. Recent studies have reported an antagonistic role of IGF signalling against the Wnt pathway[38, 39], implying a latent function of Igfbp2 in Wnt inhibition to induce Axin2$^+$-to-Tbx3$^+$ cell transition. However, these results are apparently inconsistent with a previous study demonstrating that loss of Wnt signalling (β-catenin deletion) in Axin2-expressing cells results in repression of their proliferation in the IFE basal layer[8]. This inconsistency may caused by the difference in methods to inhibit Wnt signalling because Sfrp1 attenuates Wnt ligand-receptor association, whereas β-catenin deletion suppresses the canonical Wnt pathway. It would be interesting to examine whether Sfrp1 induces the generation of Tbx3$^+$ basal cells through the non-canonical Wnt pathway. Also, we cannot exclude the possibility that Sfrp1 and Igfbp2 regulate Axin2$^+$ IFE basal cells indirectly through other cell populations in dermis.

How the dermis-derived Sfrp1/Igfbp2 signals are upregulated during pregnancy remains unknown. Sfrp1 and Igfbp2 are expressed in α-SMA$^+$/vimentin$^+$ dermal cells, implying the involvement of myofibroblasts or vascular smooth muscle cells. In this context, it is noteworthy that, in generically obese (ob/ob) mice, whose abdominal circumference was increased during the course of obesity to the same level as pregnant mice at 16 dpc (Supplementary Fig. 5B), the populations of α-SMA$^+$ dermal fibroblasts and Ki67$^+$ and Tbx3$^+$ IFE basal cells in their abdominal skin were decreased compared with wild-type mice (Supplementary Fig. 5C–E). This result suggests that skin cell dynamicity in obese mice is different from that in the abdominal skin of pregnant mice. It may be possible that tension applied on the skin or systemic hormonal regulation, which might be different between pregnancy and obesity, functions as a trigger for myofibroblast differentiation or vascular smooth muscle cell activation in the abdominal skin of pregnant mice.

The mechanism that we have described explains how IFE stem cells adapt to rapid skin expansion during pregnancy, and illustrates the similarity of IFE basal cell dynamicity between pregnancy and regeneration. Because enforcement of IFE basal cell dynamicity by administration of Sfrp1 and Igfbp2 proteins promotes wound healing, and stem cells activate in various organs during pregnancy[40, 41], these pregnancy-associated stem cell regulatory mechanisms could be applied to tissue engineering and regenerative medicine.

## Methods

**Animals and genetic experiments.** Mice were maintained on a C57BL/6 genetic background. The 8–12-week-old male mice were used for wound healing assay in Fig. 5. All other experiments were performed using 8–12-week-old female mice. *Fucci2* mice were obtained from RIKEN-Center for Developmental Biology (CDB) (CDB0203T)[16]. *Tbx3*$^{floxed/floxed}$ mice[18] (a kind gift from Anne Moon, Weis Center for Research, University of Utah) were interbred with *K14-CreERT*[20] (Jackson Laboratory, 005107). *Axin2-d2EGFP* mice were obtained from MMRRC (15749)[23].

*Axin2-Cre-ERT2* (Jackson Laboratory, 018867) and *K14-CreERT2*[27] (a kind gift from Pierre Chambon, GIE-CERBM (IGBMC)) mice were interbred with *Rosa26-H2B-GFP* mice (RIKEN-CDB, CDB0203K)[24]. *Sfrp1*$^{-/-}$ mice were obtained from RIKEN-Center for Developmental Biology (CDB) (CDB0075K)[36]. *ob/ob* mice (C57BL/6 J Ham Slc-*ob/ob*) were purchased from SLC Inc. To induce Tbx3 depletion in abdominal skin of pregnant mice, *Tbx3*$^{flox/flox}$/K14-CreERT mice were treated topically with 1 mg 4-hydroxytamoxifen (Sigma) solved in 100 μl ethanol on abdominal skin once per day during gestation from 8–15 dpc. For cell division pattern analyses, *Axin2-Cre-ERT2/Rosa26-H2B-GFP* and *K14-CreERT2/Rosa26-H2B-GFP* mice were intraperitoneally injected with tamoxifen at concentrations of 3 and 0.2 mg/25 g body weight, respectively, to induce Cre-mediated recombination. At 2 days (for *Axin2-Cre-ERT2/Rosa26-H2B-GFP* mice) or 4 days (for *K14Cre-CreERT2/Rosa26-H2B-GFP* mice) after tamoxifen administration, the mice were sacrificed and subjected to immunohistochemical analyses. For wound healing assay, 4-hydroxytamoxifen (1 mg) was topically applied to dorsal skin of *Tbx3*$^{flox/flox}$/K14-CreERT mice once per day for 5 days before induction of wounding. All experiments were performed in accordance with the guidelines of the Kyoto University Regulation on Animal Experimentation. The animal experiments in this study were approved by the Committee for Animal Experiments of the Institute for Frontier Life and Medical Sciences, Kyoto University.

The sample size was chosen by experimental consideration and not by a statistic methods. The experiments were not randomized. The investigators were not blinded to allocation during experiments and outcome assessment.

**Administration of EdU, inhibitors and recombinant proteins.** EdU (Sigma, 1 mg) was intraperitoneally injected into mice twice at 12 h intervals at 1 day before sacrifice. DMSO, PP2 (Millipore, 1 mM), NBI31772 (TOCRIS, 0.3 mM) and WAY 316606 hydrochloride (TOCRIS, 2 mM) in 100 μl were intradermally injected into mice twice a day for 4 days. IWP-2 (Sigma, 133.33 μM), Sfrp1 (R&D Systems, 20 μg/ml) and Igfbp2 (R&D Systems, 5 μg/ml) in 100 μl were intradermally injected into mice once a day for 4 days. LY364947 (Wako, 1.84 mM) in 100 μl was intradermally injected into mice once a day for 6 days.

**Immunohistochemical analysis of mouse tissues.** The medial abdominal skin area was taken as a sample for the analysis to exclude areas where developing mammary glands are present. Skin tissues were cryoprotected in 20% sucrose/PBS and frozen in optimal cutting temperature compound. The samples were sectioned and subjected to immunostaining. For detection of Ki67 and Tbx3, tissue samples were fixed with 4% paraformaldehyde, followed by permeabilization with 0.5% Triton X-100 in TBS for 30 min at room temperature. The sections were blocked with 5% bovine serum albumin at room temperature for 1 h, incubated with primary antibodies at 4 °C overnight, washed and then incubated for 1 h with secondary antibodies (Alexa Fluor 488-conjugated or 594-conjugated goat anti-rabbit, anti-mouse, anti-rat (Molecular Probes) and Alexa Fluor 488-conjugated anti-chicken (Jackson ImmunoResearch)). The following primary antibodies were used: anti-Ki67 (rabbit, 1:500, Novus, NB600-1209), anti-Tbx3 (rabbit, 1:1000, Abcam, ab99302), anti-αSMA (mouse, 1:100, Dako, M0851), anti-vimentin (rabbit, 1:100, MBL, JM-3634-100), anti-CD71 (rat,1:100, Affymetrix, 14-0711), anti-GFP (chicken, 1:200, Abcam, ab13970), anti-survivin (rabbit, 1:400, Cell Signalling, 2808), anti-Keratin10 (mouse, 1:200, Millipore, MAB3230) and anti-α18-catenin (rat, 1:1000, a kind gift from A. Nagafuchi) antibodies.

**FISH.** The coding sequences of Sfrp1 (247–1375 bp) and Igfbp2 (783–1342 bp) were amplified from mouse skin cDNAs using the following primer pairs: Sfrp1 fw, 5′-CGGAATTCGAAGGCAGCGTGGGCAG-3′; Sfrp1 rev, 5′-CGGGATCC GTTCGCTCTAGGAACAGAG-3′; Igfbp2 fw, 5′-CGGAATTCGCCCCCTGGAA-CATCTCTACTC-3′; Igfbp2 rev, 5′-CGGGATCCTTTTTTTTTTTTTTTTTTTAGTTTTCCTTCCTTTAATC-3′. The DNA fragments were subcloned into pBluescript II SK downstream of the T7 promoter (for the sense probe) or T3 promoter (for the anti-sense probe). The vectors were linearised by digestion with BamH1 (for the sense probe) or EcoRI (for the anti-sense probe). Digoxigenin (DIG)-labelled sense and anti-sense probes for Sfrp1 and Igfbp2 were amplified using DIG labelling mix (Roche), transcription buffer (Roche) and T7 or T3 polymerases (Roche). The DIG-labelled probes were isolated using an RNeasy Micro kit (Qiagen). Frozen sections (5 μm thick) were fixed in 4% paraformaldehyde, washed three times in PBS and acetylated for 10 min. The samples were then washed in PBS, treated with 5 μg/ml ProK for 15 min and washed in PBS. After treatment with 0.3% Triton/PBS for 15 min, the samples were washed three times in PBS, followed by pre-hybridisation for 30 min in hybridisation buffer (50% formamide, 5×SSC, 5×Denhardt's solution, 250 μg/ml yeast RNA and 500 μg/ml salmon sperm DNA). Hybridisation with DIG-labelled sense/anti-sense probes was performed overnight in hybridisation buffer at 75 °C. After hybridisation, the samples were blocked with Blocking One (Nacalai) for 1 h. DIG-labelled probes were detected by incubating the samples with a fluorescein-anti-DIG antibody (mouse, 1:500, Roche Applied Science, 11333089001) in Blocking One for 1 h at room temperature.

**qRT-PCR**. Total RNA was purified with the RNeasy Micro Kit according to the manufacturer's instructions. RNA (1 µg) was reverse transcribed with random primers, and the obtained cDNA was subjected to qRT-PCR analysis using a KAPA SYBR FAST Universal qPCR Kit. The primers sequences were: *Tbx3* fw, 5′-TTGC AAAGGGTTTTCGAGAC-3′; *Tbx3* rev, 5′-TGGAGGACTCATCCGAAGTC-3′; *Thy1* fw, 5′-ATGAACCCAGCCATCAGCG; *Thy1* rev, 5′-GGGTAAGGACCTT-GATATAGG-3′; *Tgm3* fw, 5′-ACAGTGCTGCGGTGCTTGGG; *Tgm3* rev, 5′-GGG GCCTAGGTCAGTCCGCA-3′; *Krt2* fw, 5′-AGATCAAAACCCTCAACAACAAA-3′; *Krt2* rev, 5′-CTCTGACTCCTGTGAGGTCCTT-3′; *Stfa1* fw, 5′-ACCTG CCA-CACCAGAAATCC; *Stfa1* rev, 5′-ACCTG CCACACCAGAAATCC-3′; *Stfa3* fw, 5′-ATTGACGGGCTGCATCTCTTT-3′; *Stfa3* rev, 5′-CTCAAGTCGTTGCTGGA-CAA-3′; *G3PDH* fw, 5′-CATCCACTGGTGCTGCCAAGGCTGT-3′; *G3PDH* rev, 5′-ACAACCTGGTCCTCAGTGTAGCCCA-3′.

**Isolation of α6 integrin+ basal cells and flow cytometry**. Subcutaneous fat was removed from the skin with a scalpel, and the whole skin was trypsinized at 4 °C overnight to remove the dermis. The cell suspensions were filtered through strainers (70-µm, BD Falcon), collected by centrifugation (300 × *g*, 5 min) and suspended in Dulbecco's modified Eagle's medium (Ca−). Cell suspensions were incubated with an Alexa Fluor 647-labelled anti-α6 integrin (BD) antibody and propidium iodide (PI) for 30 min on ice, washed twice with PBS and subjected to flow cytometric analysis. PI was used to exclude dead cells. Cell isolation was performed on a FACSAriaII sorter.

**Isolation of dermal cells**. Subcutaneous fat was removed from the skin with a scalpel, and the whole skin was incubated in Hank's balanced salt solution (HBSS; 1.25 mM CaCl$_2$ and 0.81 mM MgSO$_4$; Sigma) containing 10% CS and dispase I (25 U/ml, Wako) for 20 min at 37 °C. Dermal cell suspensions were obtained by scraping the skin gently and incubating in HBSS containing 10% CS, collagenase I (1.25 mg/ml, Life Technology), collagenase II (0.5 mg/ml, Millipore), collagenase IV (0.5 mg, Sigma), hyaluronidase (0.1 mg/ml, Sigma) and DNase I (50 U/ml, TaKaRa) for 30 min at 37 °C under gentle shaking. The cells were then filtered through 70-µm strainers, collected by centrifugation (300 × *g*, 5 min) and washed with HBSS containing 10% CS.

**Microarray analysis**. Total RNA was isolated from FACS-purified α6 integrin+ basal cells or dermal cells using Isogen (Nippon Gene) and the RNeasy Micro kit. Reverse transcription, labelling and hybridisation were performed using a GeneChip 3′IVT Express Kit (Affymetrix) according to the manufacturer's protocols. Clustering analysis was performed using Gene Spring GX12 (Digital Biology).

**Wounding**. Wounding was induced by punching roundly (5mm in diameter) the surface of back skin with BIOPSY PUNCH (Kai medical). Isoflurane was used for anaesthetising the mice.

**Statistics and reproducibility**. All experiments with or without quantification were independently performed at least three times with different mice. The statistical analysis used for each quantification data was indicated in every Figure legend.

**Data availability**. Microarray data described in this study are available on the Gene Expression Omnibus (GEO) under accession codes GSE86158. All other relevant data are available from corresponding author on request.

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

## Acknowledgements

We thank P. Chambon, Y. Hanakawa and K. Sayama for *K14-Cre-ERT2* mice, A. Moon for *Tbx3*^flox/flox mice, and A. Nagafuchi for the anti-α18-catenin antibody. We thank T. Lechler for critical scientific discussions and technical guidance for dermatology. This work was supported by JSPS KAKENHI Grant Number 26-56 (R.I.), the Joint Usage/Research Center Program of the Institute for Virus Research, Kyoto University (T.H.), the Takeda Science Foundation (F.T.), a SUMBOR GRANT (F.T.), The Nakatomi Foundation (F.T.), and JSPS KAKENHI Grant Number 16H05368 (F.T.) and 17H05640 (F.T.).

## Author contributions

R.I. and F.T. designed the study. R.I., H.K., S.Y., Y.I., H.K., S.M., S.K., H.M. and T.H. performed the experiments. R.I., H.K., S.Y., S.M., T.H. and F.T. analysed the data and interpreted the results. R.I. and F.T. wrote the manuscript.

## Additional information

**Accession codes:** Microarray data described in this study are available on the Gene Expression Omnibus (GEO) under accession codes GSE86158.

**Competing interests:** The authors declare no competing financial interests.

