## [Peer Review File · Nature Communications]

Reviewers' Comments:

Reviewer #1:

Remarks to the Author:

The authors have now made a number of alterations to their manuscript including several changes to the text and additional experiments that improve its overall quality. The data provided now better supports the model provided in this version of the paper, that a distinct Tbx3+ population arises from the normal Axin2+ IFE basal cell population specifically in growing skin during pregnancy. The authors have also addressed previous concerns about the later fate of Tbx3+ cells by both following Axin2+ clones for periods after pregnancy and by more clearly stating that the fate of these cells cannot be conclusively determined.

The authors should also change line 75 and 76 where they state "however, it is unclear whether adult IFE stem cells undergo planar-oriented cell division to generate proliferative cell progeny, and even less is known about how the stem cells and their progeny contribute to epidermal tissue reorganization during changes in physiological body shape". This sentence is inaccurate as following previous work (Clayton Nature 2007 that the authors cite as well as Rompolas Science 2016 work which the authors fail to cite) collectively shows that a single population sustain homeostasis and that planar divisions are the only detected during adult epidermal homeostasis. Therefore the sentence above should simply state: "however, it is unclear how adult IFE stem cells contribute to epidermal tissue reorganization during changes in physiological body shape"

Overall, this manuscript now provides both insight into novel epidermal biology as well as a more clearly supported model, and I feel it is appropriate for publication in Nature Communications.

In reply to the reviewer's comments:

(Reviewer #2)

The authors have now made a number of alterations to their manuscript including several changes to the text and additional experiments that improve its overall quality. The data provided now better supports the model provided in this version of the paper, that a distinct Tbx3⁺ population arises from the normal Axin2⁺ IFE basal cell population specifically in growing skin during pregnancy. The authors have also addressed previous concerns about the later fate of Tbx3⁺ cells by both following Axin2⁺ clones for periods after pregnancy and by more clearly stating that the fate of these cells cannot be conclusively determined.

Answer:

We appreciate this reviewer's positive evaluation of our previous revision.

The authors should also change line 75 and 76 where they state "however, it is unclear whether adult IFE stem cells undergo planar-oriented cell division to generate proliferative cell progeny, and even less is known about how the stem cells and their progeny contribute to epidermal tissue reorganization during changes in physiological body shape". This sentence is inaccurate as following previous work (clayton nature 2007 that the authors cite as well as rompolas science 2016 work which the authors fail to cite) collectively shows that a single population sustain homeostasis and that planar divisions are the only detected during adult epidermal homeostasis. Therefore the sentence above should simply state: "however, it is unclear how adult IFE stem cells contribute to epidermal tissue reorganization during changes in physiological body shape"

Answer:

We have corrected the sentence to conform to the reviewer's advice, and additionally cited the following reference (Rompolas, P. *et al.* Spatiotemporal coordination of stem cell commitment during epidermal homeostasis. *Science* **352**, 1471-1474, 2016) in the introduction section (page 4).

Overall, this manuscript now provides both insight into novel epidermal biology as well

as a more clearly supported model, and I feel it is appropriate for publication in Nature Communications.

Answer:

We appreciate this reviewer's positive evaluation on this manuscript.